# Establishment of Novel Mouse Model of Dietary NASH Rapidly Progressing into Liver Cirrhosis and Tumors

**DOI:** 10.3390/cancers15143744

**Published:** 2023-07-24

**Authors:** Qianqian Zheng, Masaya Kawaguchi, Hayato Mikami, Pan Diao, Xuguang Zhang, Zhe Zhang, Takero Nakajima, Takanobu Iwadare, Takefumi Kimura, Jun Nakayama, Naoki Tanaka

**Affiliations:** 1Department of Metabolic Regulation, Shinshu University School of Medicine, Matsumoto 390-8621, Japan; 20hm125b@shinshu-u.ac.jp (Q.Z.); diaopan1@gmail.com (P.D.); 21hm118c@shinshu-u.ac.jp (X.Z.); zhangzhezz0914@163.com (Z.Z.); nakat@shinshu-u.ac.jp (T.N.); 2Oriental Yeast Co., Ltd., Itabashi, Tokyo 174-8505, Japan; kawaguchi.masaya@nisshin.com (M.K.); mikami.hayato@nisshin.com (H.M.); 3Department of Medicine, Division of Gastroenterology and Hepatology, Shinshu University School of Medicine, Matsumoto 390-8621, Japan; 22hm104g@shinshu-u.ac.jp (T.I.); t_kimura@shinshu-u.ac.jp (T.K.); 4Department of Molecular Pathology, Shinshu University School of Medicine, Matsumoto 390-8621, Japan; jnaka@shinshu-u.ac.jp; 5Department of Global Medical Research Promotion, Shinshu University Graduate School of Medicine, Matsumoto 390-8621, Japan; 6International Relations Office, Shinshu University School of Medicine, Matsumoto 390-8621, Japan; 7Research Center for Social Systems, Shinshu University, Matsumoto 390-8621, Japan

**Keywords:** non-alcoholic fatty liver disease, non-alcoholic steatohepatitis, hepatocellular carcinoma, liver cirrhosis, hepatic fibrosis, mouse model

## Abstract

**Simple Summary:**

Non-alcoholic steatohepatitis (NASH) may progress into liver cirrhosis and hepatocellular carcinoma (HCC). NASH has been recognized as a major cause of HCC, which is the third leading cause of cancer-related deaths worldwide. However, the development of drug therapy for NASH and ensuing liver cirrhosis and HCC has been limited due to the lack of reliable preclinical models of the NASH progression. Here, we developed a new diet-induced mouse model of NASH-liver cirrhosis-HCC sequence. Compared with the previous mouse model, this model demonstrated shorter occurrence of NASH, liver cirrhosis, and HCC and more similar pathological characteristics to humans. Considering the similarity to clinicopathological features of human NASH in addition to very high reproducibility, generality, and convenience, our mouse model is expected to be used for the development of novel compounds for the treatment of NASH patients.

**Abstract:**

Non-alcoholic steatohepatitis (NASH), which is the most severe manifestation of non-alcoholic fatty liver disease (NAFLD), has been recognized as a major hepatocellular carcinoma (HCC) catalyst. However, the molecular mechanism of NASH-liver fibrosis-HCC sequence remains unclear and a specific and effective treatment for NASH has not yet been established. The progress in this field depends on the availability of reliable preclinical models which show the steady progression to NASH, liver cirrhosis, and HCC. However, most of the NASH mouse models that have been described to date develop NASH generally for more than 24 weeks and there is an uncertainty of HCC development. To overcome such shortcomings of experimental NASH studies, we established a novel NASH-HCC mouse model with very high reproducibility, generality, and convenience. We treated male C57BL/6J mice with a newly developed choline-deficient and methionine-restricted high-fat diet, named OYC-NASH2 diet, for 60 weeks. Treatment of OYC-NASH2 diet for 3 weeks revealed marked steatosis, lobular inflammation, and fibrosis, histologically diagnosed as NASH. Liver cirrhosis was observed in all mice with 48-week treatment. Liver nodules emerged at 12 weeks of the treatment, > 2 mm diameter liver tumors developed in all mice at 24 weeks of the treatment and HCC appeared after 36-week treatment. In conclusion, our rapidly progressive and highly reproducible NASH-liver cirrhosis-HCC model is helpful for preclinical development and research on the pathogenesis of human NAFLD-NASH-HCC. Our mouse model would be useful for the development of novel chemicals for NASH-HCC-targeted therapies.

## 1. Introduction

Hepatocellular carcinoma (HCC) is one of the most common and fatal cancers in the world [1,2,3]. In developed countries, the incidence rate of HCC has increased 3-fold in the past 30 years, becoming the fastest rising cause of cancer-related deaths [3,4,5]. Several risk factors can cause it, including hepatitis B virus (HBV), hepatitis C virus (HCV), excessive ethanol consumption, smoking, obesity, diabetes, aflatoxin, autoimmunity, hepatic iron or copper accumulation, etc. HCC was originally attributed to the emergence and spread of HBV and HCV [6]. However, in recent years, due to excessive calorie intake and sedentary lifestyles, excess lipids are stored in hepatocytes, leading to obesity and metabolic syndrome, which are increasing in clinical importance [7]. Non-alcoholic fatty liver disease (NAFLD) is related to these lifestyle changes. It has been calculated that between 4% and 22% of HCC cases can be attributed to NAFLD [8]. Currently, NAFLD is one of the leading causes of the increase in persistent liver abnormalities worldwide. It is estimated that at least 30% of adults in developed countries suffer from NAFLD, with up to 15% of these patients exhibiting some degree of non-alcoholic steatohepatitis (NASH), which is the most severe manifestation of NAFLD. In Japan, about 9–30% of Japanese people are found to have NAFLD during their annual health checkups [9]. According to the report by Eslam et al., approximately 40% of NAFLD patients suffer from progressive liver fibrosis and in this group, 20% of patients rapidly progress to advanced fibrosis or cirrhosis, a severe precancerous condition that may require liver transplantation or lead to liver failure [10].

According to liver histology, NAFLD is divided into simple steatosis and NASH, leading to liver fibrosis/cirrhosis, HCC, and ultimately death [11,12,13,14,15]. NASH is defined as the presence of hepatocyte degeneration, hepatic inflammation and/or fibrosis in addition to hepatic steatosis. Currently, NASH is the second highest cause for orthotopic liver transplantation and has become the main indication for liver transplantation in developing countries [11,16,17,18]. However, the molecular mechanisms of NAFLD-NASH-HCC progression are not fully understood. Understanding the pathogenesis of NAFLD/NASH is essential for developing new therapeutic intervention strategies. Advances in this field depend on reliable preclinical models. Therefore, it is crucial to develop a reliable mouse model of NAFLD-NASH-HCC.

Currently, NASH mouse models can be divided into four categories: (1) diet-induced models, (2) chemical-induced models, (3) gene-editing models, (4) a combination of the preceding two or three methods [19]. For evaluating the drug efficiency using NASH-HCC animal models, rapid, easy, and absolute occurrence of NASH, liver fibrosis, and liver tumor is mandatory [20,21]. However, a simple high-fat diet (HFD) typically develops dyslipidemia, fatty liver, and mild NASH without appreciable fibrosis and HCC [4,22,23,24]. 7,12-Dimethylbenz[a]anthracene injection to HFD can develop HCC after 1 year, but liver fibrosis is rare [25]. Furthermore, a mouse model involving feeding mice a choline-deficient L-amino acid-defined high-fat diet (CDAAHFD) developed steatosis, steatohepatitis, and liver fibrosis more rapidly and more severely than traditional models, as reported by Matsumoto et al., but the process was slow (NASH occurring after 24 weeks and the occurrence of liver tumors after 36 weeks) [26,27]. In addition, several specific mouse knockout models exist for studying the development of HCC in NAFLD, including mutations of genes encoding phosphatase and tensin homolog, augmenter of liver regeneration, and melanocortin 4 receptor [28,29,30]. All these models demonstrate that the formation time of HCC is more than 70 weeks. Due to these shortcomings, rapid, easy, and convenient NASH-HCC modeling is urgently needed.

In the current study, we created a novel mouse model of NASH-HCC similar to the natural course of human pathology, with very high reproducibility, generality, and convenience.

## 2. Materials and Methods

### 2.1. Animals and Experimental Design

Specific pathogen-free male C57BL/6J mice were purchased from Charles River Laboratories Japan Inc. (Yokohama, Kanagawa, Japan) and maintained with an AIN93M regular chow and tap water ad libitum. Oriental Yeast Co., Ltd. (Itabashi, Tokyo, Japan) developed a new choline-deficient and methionine-restricted high-fat diet named OYC-NASH2 diet. Components of AIN93M and the OYC-NASH2 diet are shown in Appendix A. All animal experiments were conducted using the methods outlined in the *Guide to the Care and Use of Experimental Animals*, approved by the School of Medicine of Shinshu University (approval number 020041). At 5 weeks old, the mice were randomly divided into two groups, and maintained under the same controlled clean environment (25 °C, 12 h light/dark cycle and tap water available ad libitum) [13]. One group (*n* = 41) was fed an AIN93M diet as a control group. The other group (*n* = 74, two mice died in the period of 60 weeks) was fed the OYC-NASH2 diet. Samples were taken at 0, 3, 6, 9, 12, 24, 36, 48, and 60 weeks after the feeding. All mice were maintained under the abovementioned conditions for 0–60 weeks. At each sampling point (0, 3, 6, 9, 12, 24, 36, 48, and 60 weeks), mice were weighed and then sacrificed by isoflurane anesthesia. Blood samples were collected and maintained at −80 °C until assayed. The liver was quickly removed and weighed. To assess the extent of macroscopic liver nodules (diameter > 2 mm), we scored each liver on a scale of 0–3 as follows: stage 0 shows no visible nodules on the liver; stage 1, one to three nodules; stage 2, four to six nodules; and stage 3, seven or more nodules. After that, one part of the non-tumorous liver tissue was snap-frozen on dry ice for mRNA analysis, and another small piece of non-tumorous tissue was immediately fixed in 10% neutral-buffered formalin (FUJIFILM Wako Pure Chemical Corporation, Osaka, Japan) for further histological analysis. 

### 2.2. Biochemical Analysis of Serum and Liver

Serum alanine aminotransferase (ALT), aspartate aminotransferase (AST), triglycerides (TG), total cholesterol (T-Chol), non-esterified fatty acids (NEFA), phospholipids (PL), total bile acid (TBA), and glucose were measured using a commercial enzyme assay kit (FUJIFILM Wako Pure Chemical Corporation). Total liver lipids were extracted according to the hexane: isopropanol method and quantified with a commercial assay kit (FUJIFILM Wako Pure Chemical Corporation) [31]. To determine hepatic hydroxyproline contents, frozen liver tissues (~10 mg) were hydrolyzed in 6 M HCl at 95 °C for 20 h, cooled to room temperature, and centrifuged at 13,000× *g* for 10 min. The supernatant was collected, diluted with deionized water to 4 M HCl, and subsequently diluted five-fold with 4 M HCl to avoid matrix effects [32]. Hydroxyproline measurements were performed using the QuickZyme Hydroxyproline Assay Kit (QuickZyme BioSciences, Leiden, The Netherlands) following the manufacturer’s protocol.

### 2.3. Histological Analysis

Small pieces of liver tissue were fixed in 10% formalin and embedded in paraffin. Sections of 3 μm in thickness were stained with hematoxylin and eosin (HE) or Azan–Mallory staining using standard methods. We performed a semiquantitative histological assessment of the following items with ×100, ×200 and ×400 magnifications: steatosis (score 0–3), lobular inflammation (score 0–3), portal inflammation (score 0–2), the prevalence of foam cells (score 0–2) and fibrosis (score 0–4). Steatosis score was defined as low to moderate power assessment of parenchymal involvement of steatosis. A score of 0 is given for <5%, 5–33% is 1, >33–66% is 2, and >66% is 3. Lobular inflammation was defined as overall assessment of all inflammatory foci. No foci are 0, <2 foci per 200× field is 1, 2–4 foci per 200× field is 2, >4 foci per 200× field is 3. Portal inflammation was assessed from low magnification. None is 0; minimal is 1; greater than minimal is 2. Foam cells, also called as lipid-laden macrophages, were scored since we did not observe ballooning of hepatocytes. None is 0, 1 is few foam cells, 2 is many/prominent foam cells. Azan–Mallory staining was used to observe fibrosis: none is 0; perisinusoidal or periportal is 1; perisinusoidal and portal/periportal is 2; bridging fibrosis is 3; cirrhosis is 4. These criteria were derived from the NAFLD histological scoring system proposed by Kliner et al. [33].

### 2.4. Quantification of mRNA Levels

Total RNA was extracted from frozen whole liver tissue using the RNeasy Mini kit (Qiagen, Tokyo, Japan). The RNA samples were then reverse transcribed using the Prime Script RT Reagent Kit (Perfect Real Time, Takara Bio Inc., Shiga, Japan). All mRNA levels were determined by real-time quantitative polymerase chain reaction (qPCR) on a Thermo Fisher Quant Studio 3 Real-Time PCR Instrument (Thermo Fisher Scientific, Waltham, MA, USA) using SYBR qPCR mix (Toyobo Co., Ltd., Osaka, Japan). The mRNA levels were normalized to 18S ribosomal RNA (18S rRNA) levels and expressed as change relative to the control C57BL/6J mice at 0 weeks. Primer sequences are listed in Appendix A.

### 2.5. Statistical Analysis

Results are expressed as the mean ± standard error of the mean (SEM). The two-tailed Student’s *t*-test and chi-squared test were conducted for quantitative and qualitative data, respectively, using SPSS statistics, version 22 (IBM, Armonk, NY, USA). A *p* value ≤ 0.05 was considered statistically significant.

## 3. Results

### 3.1. OYC-NASH2 Diet Feeding for 60 Weeks Results in Increased Liver Weight without Involving Severe Loss of Body Weight

Male C57BL/6J mice were treated with OYC-NASH2 diet or purified control AIN93M diet for 60 weeks. All mice were reared from 0 to 60 weeks. The body weight of mice fed the OYC-NASH2 diet was slightly less than control mice fed AIN93M, but not severely (Appendix A). The body weights of the two groups of mice tended to increase with the feeding period. At the same time, the ratio of liver to body weight in both groups of mice increased with the feeding period, reaching a maximum at 60 weeks, and the ratio of liver to body weight of mice fed the OYC-NASH2 diet was higher than the control mice fed AIN93M. The ratio of spleen to body weight in mice fed the OYC-NASH2 diet was significantly higher than that in mice fed AIN93M, with an increasing trend (3–36 weeks) and a slight decreasing trend after 36 weeks. Long-term feeding of the OYC-NASH2 diet induced sustained liver enlargement without significant weight loss. The epididymal white adipose tissue (eWAT)/body weight ratio and cecal body weight ratio did not change significantly (Appendix A). 

### 3.2. OYC-NASH2 Diet Induces the Appearance of Nodules in the Liver of Mice at 12 Weeks, and the Formation of Multiple Nodules in All Mice at 24 Weeks

From the macroscopic appearance of the mouse liver (Figure 1A), it was seen that one nodule appeared in the liver of the mice fed the OYC-NASH2 diet at 12 weeks, and after 24 weeks, multiple nodules filled the liver. We measured the number and size of nodules. The nodules were divided into three categories: small nodules (2–5 mm in diameter), medium-sized nodules (5–8 mm in diameter), and large nodules (>8 mm in diameter). Figure 1B demonstrates the proportion of mice that developed nodules on the OYC-NASH2 diet. In mice fed the OYC-NASH2 diet, no nodules were observed at 3–9 weeks, and 1 nodule (2–5 mm in diameter) was observed in one mouse (1/10) at 12 weeks; after that, nodules developed in all mice at 24 weeks (10/10). Figure 1C shows the average number of nodules at different nodule sizes in all mice fed the OYC-NASH2 diet in each group. The number of small nodules (2–5 mm in diameter) and large nodules (>8 mm in diameter) gradually increased with feeding period. The number of medium-sized nodules (5–8 mm in diameter) decreased slightly at 48 weeks and increased again at 60 weeks. The diversity of liver nodules increased from 24 weeks to 60 weeks. No nodules were observed on livers of control mice fed the AIN93M diet during the feeding period.

We stained the tumors of mice fed the OYC-NASH2 diet at 24–60 weeks with the HE method (Figure 1D). HE-stained tumors at 24 weeks of the treatment revealed increased numbers of hepatocytes with large fat droplets and structural distortion, such as narrowing of sinusoids, which was diagnosed as hepatocellular adenoma. At 36, 48, and 60 weeks, some tumors demonstrated disruption of normal liver structure showing a pseudoglandular pattern and the aberrant appearance of hepatocytes, such as giant cells, increased mitosis, prominent nucleoli, and abnormal cytoplasmic eosinophilic inclusion bodies, which were diagnosed as HCC. Notably, HCC began to appear after 36 weeks of the treatment and exhibited a “carcinoma in adenoma” pattern. (Figure 1D).

### 3.3. OYC-NASH2 Diet Induces NASH and Liver Fibrosis at 3 Weeks

In mice fed the OYC-NASH2 diet, more abundant macrovesicular lipid droplets were detected throughout the slices. Figure 2A shows a representative photomicrograph of HE-stained liver tissue in mice fed a control AIN93M diet and OYC-NASH2 diet for 3–60 weeks. Severe macrovesicular steatosis was observed in the livers of mice fed the OYC-NASH2 diet for 3–6 weeks, and the steatosis gradually diminished after 24 weeks. Lobular inflammation began to appear in mice fed the OYC-NASH2 diet for 3 weeks, gradually worsened, peaked at 12 weeks and weakened at 24 weeks, which presumably related to the progression of fibrosis. Although conventional ballooned hepatocytes were not observed in the liver tissue of mice fed the OYC-NASH2 diet, foam cells, i.e., lipid-laden macrophages, were detected at 9 weeks and increased at 24 weeks. Portal inflammation was observed in the liver tissue of mice fed the OYC-NASH2 diet for 24 weeks with score 1 (Figure 2B).

Prior to the emergence of multiple liver tumors, all mice fed the OYC-NASH2 diet exhibited moderate-to-severe fibrosis in the liver (Figure 3A). Perisinusoidal and pericentral fibrosis appeared in mice fed the OYC-NASH2 diet for 3 weeks, and the grade of fibrosis was gradually increased and reached score 4 (cirrhosis) at 36 weeks. All mice fed the OYC-NASH2 diet had liver cirrhosis at 48 weeks (Figure 3B). The progress of liver fibrogenesis was also evaluated by biochemical analyses. Hepatic hydroxyproline contents were significantly higher at 9 weeks of feeding the OYC-NASH2 diet and remained high after 9 weeks of feeding (Figure 3C). Additionally, the mRNA levels of genes encoding transforming growth factor beta 1 (*Tgfb1*), actin α-2 smooth muscle aorta (*Acta2*) and collagen type I alpha 1 chain (*Col1a1*) were significantly increased in mice fed the OYC-NASH2 diet for 3 weeks and maintained high expression levels compared with mice fed the AIN93M throughout the feeding period (Figure 3D). These results were in accordance with the pathological scores (Figure 3B).

### 3.4. Time Course Changes in Biochemical Parameters of Serum/Liver

We detected continuous changes in various biochemical parameters in the serum and liver tissues of mice fed either the AIN93M diet or the OYC-NASH2 diet during the feeding period (Figure 4). In mice fed the OYC-NASH2 diet, serum AST and ALT levels were significantly increased at 3 weeks compared with those of mice fed the AIN93M diet and remained high throughout the feeding period.

Serum T-Chol levels in the OYC-NASH2 group were slightly lower from 3 weeks to 24 weeks, but higher from 36 weeks to 60 weeks compared with the control group. In addition, the serum TBA levels in the OYC-NASH2 group continued to increase from 3 weeks and were significantly higher than control group. However, the serum glucose levels in the OYC-NASH2 group were lower than the control group during the feeding period (Figure 4A). Serum TG, NEFA and PL were not altered by the OYC-NASH2 diet.

Liver contents of T-Chol, TG, and NEFA started to increase at 3 weeks and were significantly higher than the control group throughout the feeding period (Figure 4B). These results suggest that the OYC-NASH2 diet enhances accumulation of TG and T-Chol in the liver, leading to steatosis and ensuing lipotoxicity, hepatic inflammation, fibrosis, and tumor development.

### 3.5. Time Course Changes in the Expression of NASH-HCC-Related Genes

To further clarify the molecular changes in the OYC-NASH2 diet mouse livers, the mRNA levels of genes responsible for hepatic inflammation and cellular stress were determined. The expression of proinflammatory genes encoding tumor necrosis factor alpha (*Tnf*), chemokine (C-C motif) ligand 2 (*Ccl2*), galectin 3 (*Lgals3*), and caspase 1 (*Casp1*) increased significantly at 3 weeks (Figure 5A). These proinflammatory molecules are reportedly induced by reactive oxygen species (ROS) and endoplasmic reticulum (ER) stress [34]. The mRNA levels of genes encoding ROS-producing enzymes, such as *p47phox* (*Ncf1*), and ER stress-inducible genes encoding DNA damage-inducible transcript 3 (*Ddit3*, also known as CHOP) were increased by the OYC-NASH2 diet (Figure 5B). It was documented that stress-induced overexpression of p62 (*Sqstm1*) and nuclear factor erythroid derived 2 like 2 (*Nrf2*) induced the expression of myelocytomatosis oncogene (*Myc*), leading to hepatocarcinogenesis [35]. The mRNA levels of *Sqstm1* and *Nrf2* were increased significantly even at 3 weeks of the treatment (Figure 5B). Accordingly, the mRNA levels of *Myc* and the other genes related to cell cycle regulation, such as cyclin D1 (*Ccnd1*), cyclin-dependent kinase 4 (*Cdk4*) and cyclin-dependent kinase inhibitor 2A (*Cdkn2a*), and proliferating cell nuclear antigen (*Pcna*) were similarly elevated in mice fed the OYC-NASH2 diet. The mRNA expression of α-fetoprotein (*Afp*), a conventional indicator of HCC, was elevated by the OYC-NASH2 diet (Figure 5C). These results show that a continuous feeding of OYC-NASH2 diet drives NASH and ensuing hepatocarcinogenesis in mice.

### 3.6. Time Course Changes in the Expression of Immune Cell-Related Genes

It is well recognized that in humans, macrophages, Kupffer cells (KCs), T cells, and invariant natural killer T (iNKT) cells are major drivers to progression to NASH, liver fibrosis, and HCC, but their roles vary during the progression [36]. Since the long-term OYC-NASH2 diet feeding reproduced NASH-liver cirrhosis-HCC sequence, mimicking the clinical course of NASH patients, we tried to address whether the changes in accumulation of various immune cells into the liver are also similar to those observed in the NASH patients and measured the mRNA levels of immune cell-specific genes using qPCR analysis. The expression of *Cd68* and integrin alpha M (*Itgam*), conventional genes mainly expressed in macrophages, were elevated in mice fed the OYC-NASH2 diet during the entire period (Figure 6A), corroborating the significant role of macrophages in NASH development. Recent studies revealed that embryonically-derived KCs are phenotypically converted to lipid-associated ones during fibrosis progression in human NASH [37,38]. Indeed, the expression of T cell immunoglobulin and mucin domain containing 4 (*Timd4*) and C-type lectin domain family 4, member f (*Clec4f*), which are predominantly expressed in embryonically-derived KCs, decreased after 24 weeks of feeding on the OYC-NASH2 diet (Figure 6B). On the other hand, lipid-associated KCs/macrophages highly express osteopontin which is encoded by *Spp1* gene. The *Spp1* mRNA levels were increased in OYC-NASH2 diet groups during the whole period (Figure 6B). Additionally, the mRNA levels of genes related to T cells, such as C-X-C motif chemokine receptor 6 and 3 (*Cxcr6* and *Cxcr3*, respectively) and iNKT cells, such as killer cell lectin-like receptor subfamily B member 1C (*Klrb1c*) and integrin subunit alpha 2 (*Itga2*), were significantly increased in OYC-NASH2 diet-fed mice (Figure 6C). These findings suggest the similarity in the prevalence of infiltrating immune cells during NASH-liver fibrosis-HCC progression between humans and this OYC-NASH2 model.

## 4. Discussion

In this study, we demonstrated a novel NASH-HCC-driven mouse model closely mimicking the natural course of NAFLD/NASH in humans. When we fed male C57BL/6J mice the OYC-NASH2 diet, the mice started to develop tumors in the liver at 12 weeks, and at 24 weeks the incidence of liver tumors in the mice reached 100% (Figure 1A,B). From 24 weeks to 36 weeks, various sizes of tumors were observed in all mice and the percentage of large tumors was gradually increased. HCC was histologically detected at 36-week or longer treatment. At 3 weeks of feeding, mice fed the OYC-NASH2 diet began to develop severe hepatic steatosis, which progressed to steatohepatitis at 6 weeks. Fibrosis developed rapidly, with mild fibrosis occurring at 3 weeks and reaching cirrhosis at 48 weeks. Foam cells were observed in mice fed the OYC-NASH2 diet at 9 weeks, which gradually increased to score 2 at 24 weeks. The mouse model we developed could solve the problems of long-term period of liver fibrosis and the uncertainty of HCC development in previous mouse models. Therefore, this model is likely to be more convenient to study the pathogenesis of NAFLD/NASH-related HCC and assess the usefulness of novel drugs targeting NASH-HCC.

At present, many NASH-HCC models have been reported, but few of them replicate the whole human phenotype. Most of the published studies used inherited leptin deficient mice (*ob*/*ob*), leptin resistant mice (*db*/*db*), or methionine/choline-deficient diet (MCD) mice [39]. The *ob*/*ob* and *db*/*db* mice do not spontaneously develop NASH and HCC. The MCD model is one of the most used tools in NASH research. One disadvantage of MCD-NASH is that mice show significant body weight loss and develop HCC spontaneously [40,41]. The CDAAHFD model has been proved to be able to simulate human NASH, liver fibrosis and HCC in mice and rats without any weight loss [42,43,44,45], but it takes more than 24 weeks to develop liver nodules and NASH. These mouse models have the same problem with respect to long-term treatment for NASH-HCC development and low prevalence of liver tumors and HCC. However, we solved the above problems by feeding C57BL/6J mice with an OYC-NASH2 diet, making the time shorter to induce NASH (at 3 weeks) and gradually develop into HCC at 36 weeks. In addition, for the OYC-NASH2 diet, we designed the types and mixing proportion of raw materials to enhance the robustness of the feed, to stabilize the palatability for test animals and reduce the variability of pathological data. The OYC-NASH2 diet can also be easily stored at room temperature (no freezer or refrigerator is required for storage), making this model more convenient.

NASH is expected to become the main cause of global liver cirrhosis and HCC. However, there are currently no approved drugs for treating NASH. Unless the pathogenesis is precisely understood, its treatment can hardly make progress. The OYC-NASH2 diet increased cholesterol in the liver of mice (Figure 4B). We already know that as the main lipotoxic molecule, liver free cholesterol (FC) promotes the development of progressive liver inflammation and fibrosis in patients with NAFLD, leading to NASH and even progression to cirrhosis and HCC [3]. Hepatic T-Chol levels in mice fed the OYC-NASH2 diet increased at 3 weeks of feeding and remained high throughout the feeding period. Elevated cholesterol can activate KCs to release inflammatory factors. Mice fed with the OYC-NASH2 diet for 3 weeks showed severe hepatic steatosis and inflammatory cell infiltration, and the expression of its related genes, such as *Tnf*, *Ccl2*, and *Lgals3*, was significantly increased at 3 weeks. Cholesterol-induced activation of KCs can lead to the activation of hepatic stellate cells (HSCs) through the release of proinflammatory cytokines, resulting in the development of fibrosis [46]. The expression of fibrosis-related genes, such as *Tgfb1*, *Acta2*, and *Col1a1*, were significantly elevated at 3 weeks of feeding on the OYC-NASH2 diet (Figure 3C). Recently, it has been proved that the accumulation of FC in HSCs can directly activate HSCs into fibroblasts and this process may play a role through the toll like receptor 4-dependent pathway [47,48]. Therefore, we speculate that the formation of NASH is related to the increase of cholesterol. The liver histopathology in mice fed with the OYC-NASH2 diet showed many foam cells at 12 weeks. It has been found that KCs can surround and process fatty dead hepatocytes containing cholesterol crystals, and then transform them into activated foam cells containing FC and cholesterol crystals [49,50]. In addition, low density lipoprotein molecules are oxidized and absorbed by macrophages to become congested and form foam cells. These foam cells often become trapped in the walls of blood vessels and contribute to atherosclerotic plaque formation [51,52,53]. According to the lipid hypothesis, elevated cholesterol levels in the blood can lead to atherosclerosis, which may increase the risk of heart attack, stroke and peripheral artery disease. The expressions of genes related to oxidative stress and ER stress, such as *Ncf1*, *Ddit3*, *Sqstm1*, and *Nrf2*, were also significantly increased at 3 weeks. Additionally, *Sqstm1* and *Nrf2* were reportedly key inducers of *Myc* [35]. Collectively, oxidative stress and ER stress in livers of OYC-NASH2 diet-fed mice may be associated with aggravated hepatocyte damage, liver inflammation, fibrosis, and hepatocarcinogenesis.

In humans, there are notable changes in immune cell populations within the liver microenvironment during NASH-liver cirrhosis-HCC sequence [36]. In the early stages of NASH, macrophages and natural killer cells are activated as naive immune cells to kill tumor cells [54]. *Cd68* is highly expressed in the entire stage of NASH-liver cirrhosis-HCC, reflecting key roles of macrophages. Additionally, previous studies have shown that *Timd4* and *Clec4f* are highly expressed in embryonically-derived KCs, but in NASH, the self-maintenance of embryonically-derived KCs is impaired, and the number of monocyte-derived KCs cells with low *Timd4*/*Clef4f* expression is increased [37,55]. With lipid accumulation and sustained liver injury, lipid-associated macrophages with high expression of osteopontin were increased [38]. Increased levels of *Spp1* mRNA, encoding osteopontin, and decreased ones of *Timd4*/*Clef4f* mRNA were detected during the treatment with the OYC-NASH2 diet. Additionally, the number of T cells and iNKT cells is increased in HCC [54]. In mice fed the OYC-NASH2 diet, the mRNA levels of genes related to T cell-specific chemokine receptors, such as *Cxcr6* and *Cxcr3*, and iNKT cell markers, such as *Klrb1c* and *Itga2*, were significantly increased during the entire period. These similarities prompt us to speculate that this OYC-NASH2 model may apply to the development of immunomodulating therapies against NASH-HCC, which could translate into use for NASH patients in the future. Further studies are needed to identify immunological characteristics of the OYC-NASH2 model in more detail. 

This study also has some limitations. It is reported that insulin resistance and diabetes are associated with NAFLD [56]. In our study, the serum glucose levels were lower in OYC-NASH2 mice compared with the control mice. Studies have shown that *db*/*db* mice and *foz*/*foz* mouse models are helpful for the study of diabetes combined with NASH [39]. Further improvement is essential in order to develop more suitable mouse models mimicking the condition of NAFLD/NASH patients having obesity and/or diabetes. Perhaps the OYC-NASH2 diet combined with *db*/*db* or *foz*/*foz* mouse models can be utilized in the future.

## 5. Conclusions

The OYC-NASH2 diet-fed mouse model mimics the natural history of human NASH-liver cirrhosis-HCC with very high reproducibility, versatility, and convenience. The OYC-NASH2 model is expected to facilitate the development of new targeted therapeutic or preventive strategies for NASH-HCC.

## Figures and Tables

**Figure 1 cancers-15-03744-f001:**
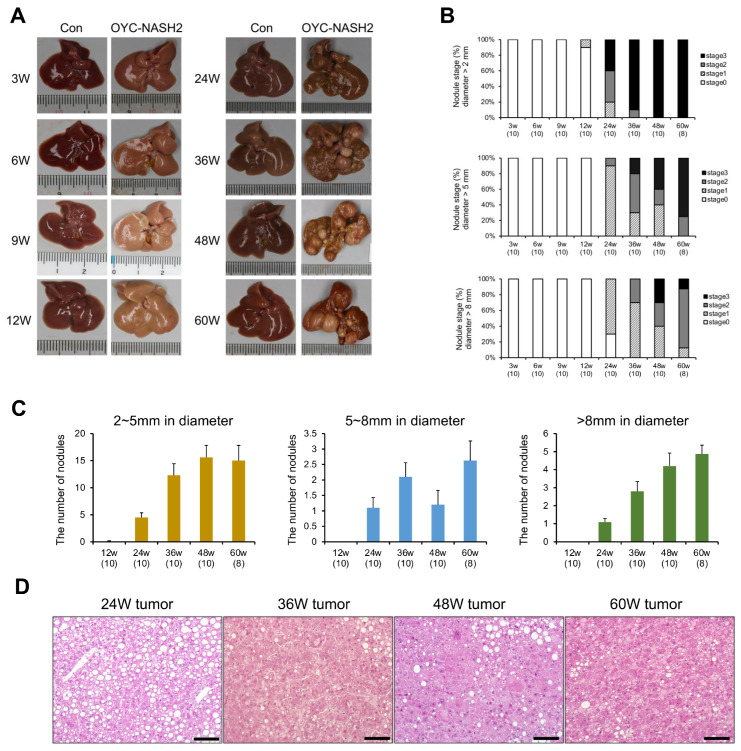
Time course of gross appearance and nodule formation in livers of C57BL/6J mice fed OYC-NASH2 diet. (**A**) Gross appearance of mouse livers. The control group (Con) fed an AIN93M diet and the OYC-NASH2 group was fed the OYC-NASH2 diet. (**B**) Proportion of mice having liver tumors. Stage 0, no visible nodules on the liver; stage 1, one to three nodules; stage 2, four to six nodules; and stage 3, seven or more nodules. (**C**) The numbers of nodules in each mice fed the OYC-NASH2 diet. Data are expressed as the means ± SEM. (**D**) Representative histopathological features of HE-stained liver tumors from mice fed the OYC-NASH2 diet for 24, 36, 48, and 60 weeks (×200 magnification). Scale bars = 50 μm. The number in parentheses in (**B**,**C**) represents the number of tested mice at each feeding period. W, week of the treatment.

**Figure 2 cancers-15-03744-f002:**
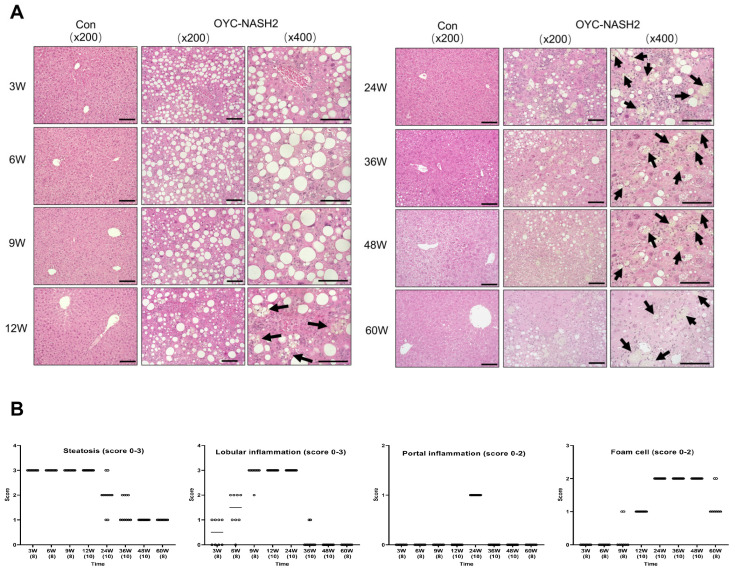
Time course of histopathological features in non-tumorous liver tissues of C57BL/6J mice fed OYC-NASH2 diet. (**A**) A representative photomicrograph of HE-stained liver tissues of OYC-NASH2 diet- (×200, ×400 magnification) or control (Con) AIN93M diet-fed (×200 magnification) mice. Scale bars = 50 μm. (**B**) Histopathological severity of NASH pathologies in OYC-NASH2 diet-fed mice. The degree of steatosis (score 0–3), lobular inflammation (score 0–3), and portal inflammation (score 0–2) and the prevalence of foam cells (score 0–2) were scored. The arrows indicate foam cells. The numbers in parentheses are the number of tested mice at each feeding period. W, week of the treatment.

**Figure 3 cancers-15-03744-f003:**
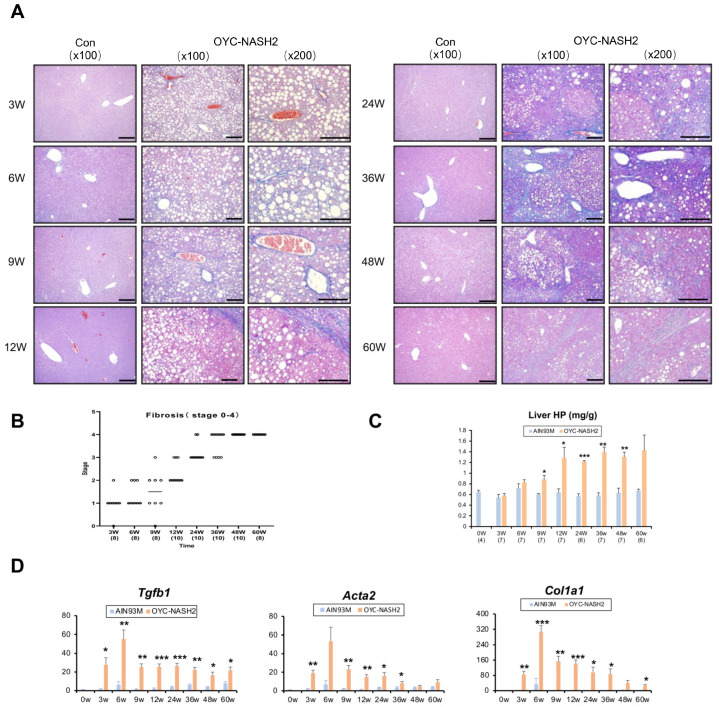
Time course of fibrogenesis in non-tumorous liver tissues of C57BL/6J mice fed OYC-NASH2 diet. (**A**) A representative photomicrograph of liver tissues stained by the Azan–Mallory method in OYC-NASH2 diet- (×100, ×200 magnification) or control (Con) AIN93M diet-fed (×100 magnification) mice at every time point. Scale bars = 50 μm. (**B**) Histopathological score of fibrosis in OYC-NASH2 diet-fed mice. The stage of fibrosis (0–4) was scored. (**C**) Hepatic hydroxyproline (HP) contents. (**D**) The mRNA levels of the *Tgfb1*, *Acta2* and *Col1a1* genes in the non-tumorous liver tissues by qPCR analysis. The numbers in parentheses are the number of tested mice at each feeding period. Data are expressed as the means ± SEM. * *p* < 0.05, ** *p* < 0.01 and *** *p* < 0.001 between AIN93M group and OYC-NASH2 group. W, week of the treatment.

**Figure 4 cancers-15-03744-f004:**
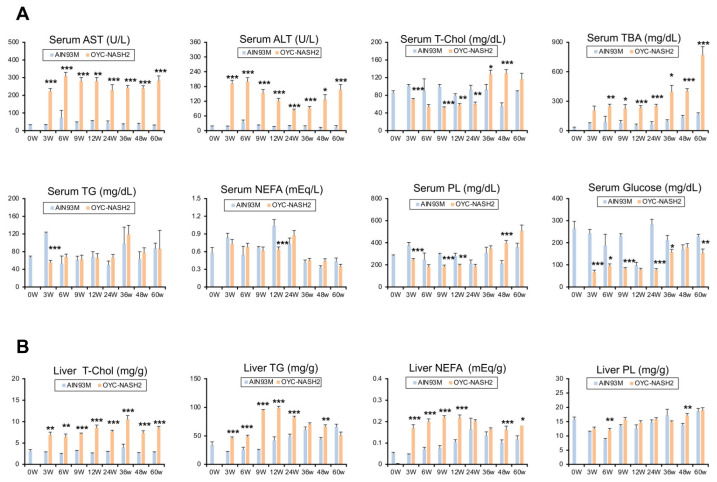
Time course of serum/liver biochemical parameters of mice fed the OYC-NASH2 diet. (**A**,**B**) Serum (**A**) and liver (**B**) biochemical parameters of mice fed the AIN93M diet or the OYC-NASH2 diet for 60 weeks. Data are expressed as the means ± SEM. * *p* < 0.05, ** *p* < 0.01 and *** *p* < 0.001 between AIN93M diet group and OYC-NASH2 group. W, week of the treatment.

**Figure 5 cancers-15-03744-f005:**
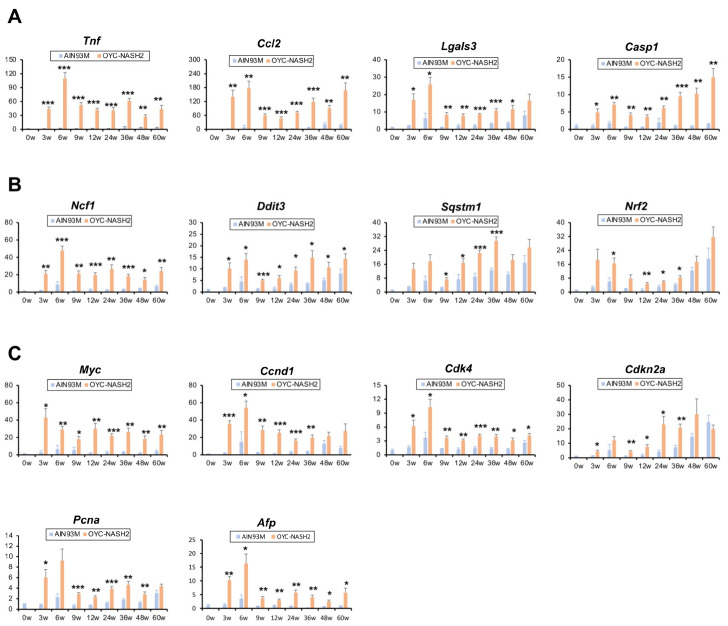
Time course of the expression of NASH-HCC-related genes in non-tumorous liver tissues of C57BL/6J mice fed OYC-NASH2 diet. The mRNA levels of genes associated with inflammation (**A**), oxidative stress and ER stress (**B**), and hepatocarcinogenesis (**C**) in non-tumorous liver tissues. The values were normalized to 18S rRNA levels and subsequently expressed as values relative to those of control diet mice at 0 weeks. Data are expressed as the means ± SEM. * *p* < 0.05, ** *p* < 0.01 and *** *p* < 0.001 between AIN93M diet group and OYC-NASH2 group. W, week of the treatment.

**Figure 6 cancers-15-03744-f006:**
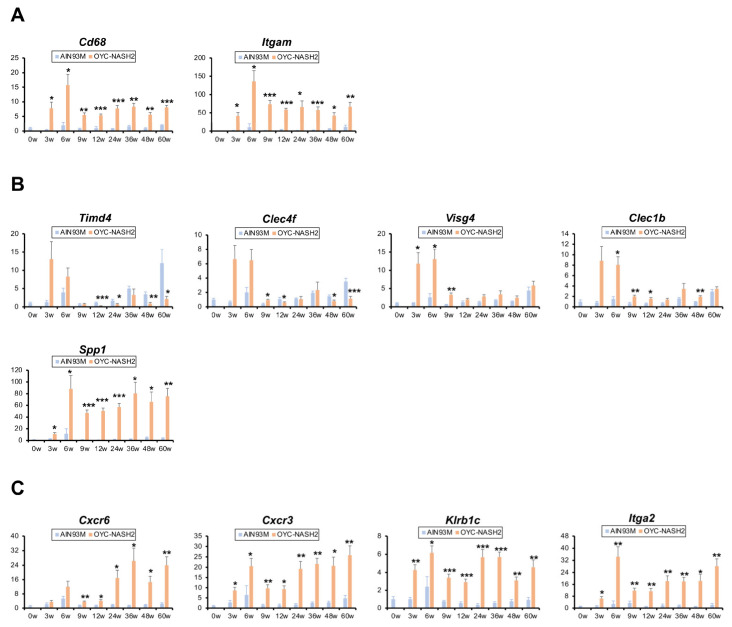
Time course of the expression of immune cell-related genes in non-tumorous liver tissues of C57BL/6J mice fed OYC-NASH2 diet. The mRNA levels of genes associated with macrophages (**A**), KCs (**B**), and T cells and iNKT cells (**C**) in non-tumorous liver tissues. The values were normalized to 18S ribosomal RNA levels and subsequently expressed as values relative to those of control diet mice at 0 weeks. Data are expressed as the means ± SEM. * *p* < 0.05, ** *p* < 0.01 and *** *p* < 0.001 between AIN93M diet group and OYC-NASH2 group. W, week of the treatment.

## Data Availability

The data presented in this study are available in this article and Appendix A.

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
