# Peer review of "Establishment of Novel Mouse Model of Dietary NASH Rapidly Progressing into Liver Cirrhosis and Tumors"

_cancers, 2023, doi:10.3390/cancers15143744_

Round 1

Reviewer 1 Report

I congratulate your excellent work. It was a very interesting study, and well written in English. 

Author Response

Reviewer 1

I congratulate your excellent work. It was a very interesting study, and well written in English.

Response

We deeply appreciate your favorable comments.

Reviewer 2 Report

In this study, the authors proposed a novel mouse model of Non-alcoholic steatohepatitis (NASH)- hepatocellular carcinoma (HCC). They treated male C57BL/6J mice with a newly developed choline-deficient and methionine-restricted high-fat diet (CDMRHFD) for 60 weeks. Treatment of CDMRHFD for 3 weeks revealed marked steatosis, lobular inflammation, and fibrosis, histologically recognized as NASH. Liver cirrhosis was observed in all mice with 36-week treatment. Liver nodules emerged at 12 weeks of the treatment, and diameter >2 mm liver tumors developed in all mice at 24 weeks of the treatment. They concluded that the proposed NASH-liver cirrhosis-HCC model is helpful for preclinical development and research on the pathogenesis of human NAFLD-NASH-HCC and that their model might be useful for the development of novel chemistry for NASH-HCC-targeted therapies or HCC prevention strategies.

The topic addressed in this study is of current major clinical interest. However, since it is well known that the disease progression of steatosis to NASH, cirrhosis, and HCC is characterized by a central role of inflammation and immune cells, and conclude that their model may be useful for preclinical development and research on the pathogenesis of human NAFLD-NASH-HCC, the authors should provide the histological features of immune cell infiltration. 

Minor editing of English language required

Author Response

Reviewer 2

In this study, the authors proposed a novel mouse model of Non-alcoholic steatohepatitis (NASH)- hepatocellular carcinoma (HCC). They treated male C57BL/6J mice with a newly developed choline-deficient and methionine-restricted high-fat diet (CDMRHFD) for 60 weeks. Treatment of CDMRHFD for 3 weeks revealed marked steatosis, lobular inflammation, and fibrosis, histologically recognized as NASH. Liver cirrhosis was observed in all mice with 36-week treatment. Liver nodules emerged at 12 weeks of the treatment, and diameter >2 mm liver tumors developed in all mice at 24 weeks of the treatment. They concluded that the proposed NASH-liver cirrhosis-HCC model is helpful for preclinical development and research on the pathogenesis of human NAFLD-NASH-HCC and that their model might be useful for the development of novel chemistry for NASH-HCC-targeted therapies or HCC prevention strategies.

The topic addressed in this study is of current major clinical interest. However, since it is well known that the disease progression of steatosis to NASH, cirrhosis, and HCC is characterized by a central role of inflammation and immune cells and conclude that their model may be useful for preclinical development and research on the pathogenesis of human NAFLD-NASH-HCC, the authors should provide the histological features of immune cell infiltration.

Response:

Thank you very much for your important suggestion. In human NASH, macrophages, Kupffer cells (KCs), T cells, and invariant natural killer T (iNK) cells are major drivers to progress into NASH, liver fibrosis, and HCC [1]. We realize that extracting infiltrating immune cells from NASH livers at every stage and characterizing immune phenotypes are the best way to answer your queries, but we cannot perform this procedure due to limited revised period. Alternatively, we determined the expression of immune cell-specific genes using qPCR analysis. The expression of Cd68 and Itgam, conventional genes mainly expressed in macrophages, was elevated in mice fed the OYC-NASH2 diet during the entire period (Figure 6A), corroborating the significant role of macrophages in NASH. Recent studies revealed that the embryonically derived KCs are phenotypically converted to lipid-associated ones during fibrosis progression in human NASH. Indeed, the expression of T cell immunoglobulin and mucin domain containing 4 (Timd4) and C-type lectin domain family 4, member f (Clec4f) was decreased after 24 weeks of feeding in OYC-NASH2 diet groups (Figure 6B). Lipid-associated macrophages highly express Spp1 encoding osteopontin. The mRNA expression of Spp1 was increased in OYC-NASH2 diet groups (Figure 6B) during the whole period. Additionally, the mRNA levels of genes related to T cells (Cxcr6 and Cxcr3) and iNK cells (Klrb1c and Itga2) were significantly increased in OYC-NASH2 diet-fed mice (Figure 6C). These findings indicate the similarity in the prevalence of infiltrating immune cells during NASH-HCC progression between humans and this mouse model. We added the abovementioned statements and related references in the Results and Discussion sections.

Reviewer 3 Report

Zheng et al reported a novel mouse model of NASH that progressed into liver cirrhosis and tumors. This may be of great interests in the field of mouse HCC models. However, major improvement must be made before further evaluation of current manuscript.

1. English language should be edited and improved by native writers.

2. All current figures must be replaced with high resolution ones. Histology features and gross appearance, as well as text labels should be visible.

3. Authors claimed that mice fed with OYC-NASH2 diets “100% developed into HCC at 24 weeks”. Unfortunately, I can’t find any histopathology showing HCC. Hepatocellular carcinoma should be confirmed with features such as trabecular, pseudoglandular or giant cell patterns, which can be found in textbooks or review papers (e.g. PMID: 25473149). Authors must at least provide representative H&E staining of claimed HCC samples. An example of such analysis can be found in many other studies, for example, PMID: 28895242 (ref 2 in current manuscript).

Minor points:
1. The introduction part needs to be re-organized. Alcohol and HBV infection, and other etiological factors should not be omitted.

2. Authors may include analysis for relevant molecules, like Tnfa, Acta2, Afp or Mki67.

English language should be edited and improved by native or highly proficient writers.

Author Response

Reviewer 3

Zheng et al reported a novel mouse model of NASH that progressed into liver cirrhosis and tumors. This may be of great interests in the field of mouse HCC models. However, major improvement must be made before further evaluation of current manuscript.

  1. English language should be edited and improved by native writers.

Response

Thank you very much for your suggestion. We fixed grammatical errors under the instruction of a native English speaker.

  1. All current figures must be replaced with high resolution ones. Histology features and gross appearance, as well as text labels should be visible.

Response

Thank you very much. We re-organized the images and improve the clarity.

  1. Authors claimed that mice fed with OYC-NASH2 diets “100% developed into HCC at 24 weeks”. Unfortunately, I can’t find any histopathology showing HCC. Hepatocellular carcinoma should be confirmed with features such as trabecular, pseudoglandular or giant cell patterns, which can be found in textbooks or review papers (e.g. PMID: 25473149). Authors must at least provide representative H&E staining of claimed HCC samples. An example of such analysis can be found in many other studies, for example, PMID: 28895242 (ref 2 in current manuscript).

Response

Thank you so much. We took photomicrographs again to present typical histological features of HCC (Figure 1D). A representative photomicrograph of tumors in 36-week feeding demonstrates trabecular HCC with pseudoglandular pattern. A photomicrograph of tumors in 48-week feeding group showed giant cell-type HCC. Finally, a representative photomicrograph in 60-week feeding showed monotonous aberrant hepatocyte proliferation with eosinophilic intracytoplasmic inclusion bodies and structural atypia. Based on these findings, HCC may develop after 36 weeks of the OYC-NASH2 diet treatment. We tempered the enthusiasm of description regarding HCC development throughout the manuscript.

Minor points:

  1. The introduction part needs to be re-organized. Alcohol and HBV infection, and other etiological factors should not be omitted.

Response

Yes, thank you very much for your suggestion. We added the descriptions in the Introduction section in more detail.

  1. Authors may include analysis for relevant molecules, like Tnfa, Acta2, Afp or Mki67.

Response

Thank you for your suggestion. We added these data to Figure 3D, 5A and 5C. Mki67 primer is not good, so we adopted PCNA (proliferating cell nuclear antigen) data instead.

[1] T. Huby, E.L. Gautier, Immune cell-mediated features of non-alcoholic steatohepatitis, Nat Rev Immunol, 22 (2022) 429-443.

Round 2

Reviewer 2 Report

The Authors provided a revised manuscript addressing the raised points.

Reviewer 3 Report

All my previous points have been addressed. I have no more major questions.

Authors may want to thoroughly examine references to make sure they are best relevant. Some of current ones may not be good. Here are some examples:

(A) Refs 1&2 are on speficic topics rather than a gross view of HCC prevalence, significance, etc.

(B) Refs 11~15 are not relevant to NAFLD classification.

(C) Refs 16~18 do not address liver transplantation.

(D) Ref 27 is not the work from Matsumoto et al.

(E) Ref 46 does not directly study KC or HSC activation.